# LG-VQ: Language-Guided Codebook Learning

**Guotao Liang[1,2], Baoquan Zhang[* 1], Yaowei Wang[2], Xutao Li[1], Yunming Ye[1], Huaibin Wang[1]**
**Chuyao Luo[1], Kola Ye[3], Linfeng Luo[3]**
[1]Harbin Institute of Technology, Shenzhen, [2]Peng Cheng Laboratory, [3]SiFar Company
{lianggt, wangyw}@pcl.ac.cn
{23B951062, 22S051022}@stu.hit.edu.cn
{baoquanzhang, lixutao, yeyunming}@hit.edu.cn
{luochuyao.dalian, kolaygm, llf10811020205}@gmail.com

## Abstract

Vector quantization (VQ) is a key technique in high-resolution and high-fidelity image synthesis, which aims to learn a codebook to encode an image with a sequence of discrete codes and then generate an image in an auto-regression manner. Although existing methods have shown superior performance, most methods prefer to learn a single-modal codebook (*e.g.*, image), resulting in suboptimal performance when the codebook is applied to multi-modal downstream tasks (*e.g.*, text-to-image, image captioning) due to the existence of modal gaps. In this paper, we propose a novel language-guided codebook learning framework, called LG-VQ, which aims to learn a codebook that can be aligned with the text to improve the performance of multi-modal downstream tasks. Specifically, we first introduce pre-trained text semantics as prior knowledge, then design two novel alignment modules (*i.e.*, Semantic Alignment Module, and Relationship Alignment Module) to transfer such prior knowledge into codes for achieving codebook text alignment. In particular, our LG-VQ method is model-agnostic, which can be easily integrated into existing VQ models. Experimental results show that our method achieves superior performance on reconstruction and various multi-modal downstream tasks.

## 1 Introduction

In recent years, with the growing development of various multi-modal task scenarios [37, 36, 38], unified modeling of visuals and language has sparked considerable interest. Vector Quantization (VQ)-based image modeling technique, exemplified by VQ-VAE [43] and VQ-GAN [9], has emerged as a pivotal approach in the realm of unified modeling. The VQ methodology [43] typically follows a two-stage generation paradigm. In the initial stage, a trainable discrete codebook is employed to quantize continuous image features into a discrete token sequence to finish the reconstruction task. Subsequently, the codebook is utilized for various downstream tasks by generative models [42, 37].

Learning a robust codebook during the initial stage is crucial for optimizing performance in downstream tasks. At present, lots of VQ methods have been proposed to achieve robust code representation [15, 14, 7, 12]. For instance, VQ-GAN [9] introduces an adversarial training loss to learn a perceptually rich codebook. Some other works consider improving the codebook representation from the perspective of addressing the problem of codebook collapse [53, 52].

Although existing methods have shown superior performance, most methods only focus on learning a single-modal codebook contains more low-level information (*e.g.*, image's pixel, edge, and texture), resulting in suboptimal performance when the codebook is applied to multi-modal downstream tasks

---

*Corresponding Authors

38th Conference on Neural Information Processing Systems (NeurIPS 2024).

(*e.g.*, text-to-image [37], image captioning [36], VQA [24]). That is because the codebook lacks high-level semantics and the existence of modal gaps.

To address the above issue, we propose a novel codebook learning method (*i.e.*, multi-modal codebook learning), called **L**anguage-**G**uided VQ (LG-VQ). The novelty lies in utilizing pre-trained text semantics as supervised information to guide the codebook to learn abundant multi-modal knowledge.

Specifically, we first employ a cross-modal pre-trained model (*i.e.*, CLIP [32]) to encode text semantics. Then, we propose two novel semantic supervision modules to transfer the text semantics into codebook, *i.e.*, Semantic Alignment Module, and Relationship Alignment Module. Within the semantic alignment module, we enhance the consistency between the semantic representations of the codebook and text through global semantic alignment and masked text prediction. On the other hand, simply aligning the text and codebook in the holistic semantic space cannot satisfy more complex reasoning tasks like image captioning and VQA. Inspired by some VQA techniques [26, 40, 24], the semantic relationships between words play a very important role in various tasks of natural language processing (See Fig. 1). Based on this fact, we further propose to transfer the semantic relationships between words into

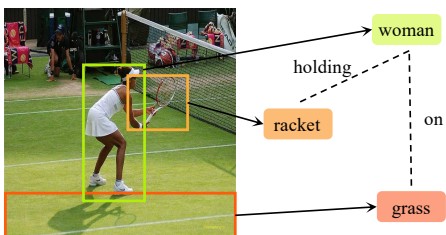

Q: What is this woman holding?

Figure 1: To answer the question, one not only needs to identify "women" and "racket" but also understand the semantic relationship between them ("holding").

codes to achieve better alignment between the codes and words. Such a text-aligned codebook helps alleviate modal gaps and improve codebook performance on cross-modal tasks.

The contributions of this work are summarized as follows:

- We point out the limitations of existing methods in learning an expressive codebook since they learn a single-modal codebook. We propose a novel multi-modal codebook learning method, named LG-VQ, which can enable the codebook to effectively retain fine-grained reconstruction information while aligning with the text.

- Resorting to pre-trained text semantics, we propose two novel semantic supervision modules, *i.e.*, Semantic Alignment Module and Relationship Alignment Module, effectively learn text-aligned codebook. The advantage of such alignment modules is the abundant context and relationship semantics contained in pre-trained text can be sufficiently leveraged for enhancing multi-modal codebook learning.

- We conduct comprehensive experiments on four public datasets, which shows that our LG-VQ method outperforms various state-of-the-art models on reconstruction and various cross-modal tasks (*e.g.*, text-to-image, image captioning, VQA).

## 2 Related Works

### 2.1 Vector Quantization for Image Generation

Vector quantization (VQ) is designed to learn a codebook, which aims to encode continuous image features into a discrete sequence. Then, the learned codebook can be utilized for various downstream tasks. Oord et al. [43] first propose a novel VQ method called VQ-VAE. This method innovatively replaces the prior distribution of Variational Autoencoder (VAE) with a discrete deterministic distribution (*i.e.*, a codebook). To further improve the performance of VQ, various models are proposed to learn a more expressive codebook [9, 48, 2, 17, 7, 15, 14, 21]. For example, VQ-GAN [9] addresses the issue of image blur generated by VQ-VAE through the introduction of an adversarial training loss. However, the above methods do not tackle the codebook collapse issue. To address the issue, many novel methods are proposed from the perspective of regularization [33], codebook update [53], codebook transfer [51]. Recently, inspired by the large language models (LLMs), instead of mapping images to the visual code tokens, some works attempt to map the images to the word tokens of LLMs by viewing images as "foreign languages" [22, 50, 56]. However, because of the inherent differences between vision and language, these works have difficulty assigning correct semantic words to images.

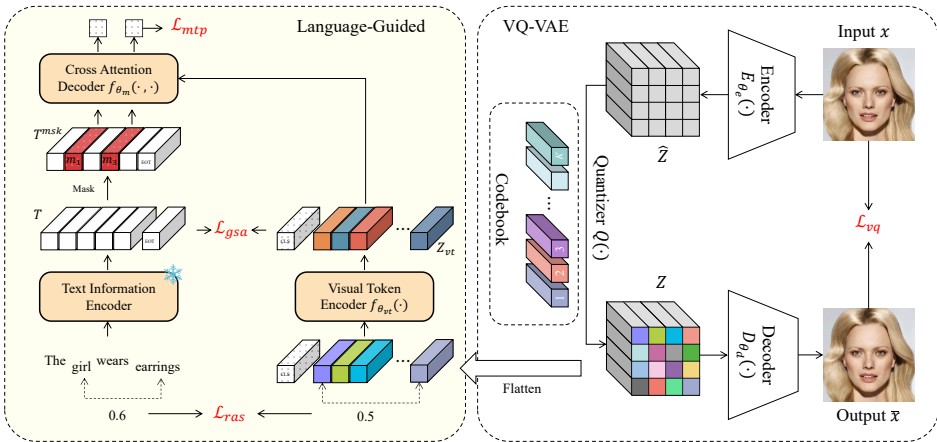

Figure 2: The overall architecture of the proposed LG-VQ method. The right part of the figure is the basic VQ-VAE module, left is our language-guided module, which consists of three losses: global semantic alignment ($\mathcal{L}_{gsa}$), masked text prediction ($\mathcal{L}_{mtp}$), and relationship alignment supervision ($\mathcal{L}_{ras}$). Here, pre-trained text information guides discrete code tokens of an image to learning rich semantic knowledge based on three losses.

Compared with the aforementioned methods, our approach focuses more on multi-modal alignment in feature space (*i.e.*, learning a text-aligned codebook). We use pre-trained text semantics to supervise the codebook learning. The advantage is that the rich semantic information from the text can be fully exploited for more robust codebook learning so that the codebook can not only retain more reconstruction information but also be able to understand and match text. More importantly, our method is model-agnostic, which can be easily integrated into existing VQ models.

## 2.2 Vision-Language Representation Learning

Vision-language Pre-training (VLP) aims to learn multi-modal representations from large-scale image-text pairs that can improve vision-language downstream tasks, for example, VQA[1]. Early methods such as LXMERT [41], UNITER [4] employ pre-trained object detectors to extract image region features, and fuse image features with text by a cross-modal encoder to achieve the vision-language representation learning. Although these methods achieve superior performance on downstream tasks, they require high-resolution input images and pre-trained object detectors. To remove the object detectors, a large number of researchers focus on learning two separate representations for image and text [32, 16, 18]. For instance, CLIP [32] learns a robust representation for each image and text using contrastive learning based on large-scale image-text pair data.

In this paper, we propose to employ pre-trained text semantics as supervised information to guide codebook learning. Its advantage is that abundant multi-modal knowledge contained in text can be fully leveraged for robust codebook learning. Additionally, we design a novel relationship alignment module to inject semantic relationships between words into codes.

## 3 Methodology

### 3.1 Preliminaries: VQ-VAE

VQ-VAE [43], as a pioneering work on the VQ research domain, aims to learn a discrete codebook to encode images into discrete token sequences through an Encoder-Decoder framework. As illustrated in Fig. 2 right, the VQ-VAE consists of a visual encoder $E_{\theta_e}(\cdot)$ with parameter $\theta_e$, a token decoder $D_{\theta_d}(\cdot)$ with parameter $\theta_d$, a quantizer $Q(\cdot)$, and a codebook is defined as $\mathcal{Z} = \{e_k\}_{k=1}^{K}$ that consists of learnable $K$ entries $e_k \in \mathbb{R}^{d_z}$ with dimension $d_z$. Given an input image $x \in \mathbb{R}^{H \times W \times C}$, where $H$, $W$, and $C$ represent the height, width, and channel of the image respectively. The visual encoder $E_{\theta_e}(\cdot)$ learns to convert the original image into grid features $\hat{Z} = E_{\theta_e}(x) \in \mathbb{R}^{\frac{H}{f} \times \frac{W}{f} \times d_z}$ and $f$ is the down-sampling factor. The quantizer $Q(\cdot)$ looks up the nearest neighbor in the codebook for each

grid representation $\hat{z}_i \in \mathbb{R}^{d_z}$ in $\hat{Z}_i$ using the following equation:

$$z_i = Q(\hat{z}_i) = \underset{e_k \in \mathcal{Z}}{arg\,min}\|\hat{z}_i - e_k\|. \tag{1}$$

The token decoder $D_{\theta_d}(\cdot)$ is used to reconstruct the original image by $\widetilde{x} = D_{\theta_d}(Z)$, where $Z$ is discrete code tokens of whole image obtained by Eq. 1. During training, the visual encoder $E_{\theta_e}(\cdot)$, codebook $\mathcal{Z}$, and token decoder $D_{\theta_d}(\cdot)$ are jointly optimized by minimizing the following objective:

$$\mathcal{L}_{vq} = \|x - \widetilde{x}\|_2^2 + \|sg[E_{\theta_e}(x)] - Z\|_2^2 + \omega\|E_{\theta_e}(x) - sg[Z]\|_2^2, \tag{2}$$

where, the first term is reconstruction loss, which measures the difference between the original image $x$ and the reconstructed image $\widetilde{x}$. $sg[\cdot]$ represents the stop-gradient operator, and the second term is codebook loss, which encourages the codebook to be close grid features. The third term is the "commitment loss" [43], where $\omega$ serves as a hyper-parameter. However, existing VQ-based methods mainly focus on the learning of single-modal codebook, thereby limiting their applicability to multi-modal downstream tasks.

## 3.2 Proposed Method: LG-VQ

Existing works attempt to improve codebook reconstruction capabilities to obtain better performance on downstream tasks. However, ignoring modal differences results in suboptimal performance when the codebook is applied to cross-modal tasks. To address this issue, we propose to utilize the pre-trained text semantics as supervised information to learn a text-aligned codebook. Its advantage is abundant semantic information from text can be fully exploited for more robust codebook learning to improve the performance of reconstruction and cross-modal tasks. The comprehensive architecture of the proposed LG-VQ method is illustrated in Fig. 2 left. It consists of two supervision modules: Semantic Alignment Module (*i.e.*, $\mathcal{L}_{gsa}$ and $\mathcal{L}_{mtp}$), and Relationship Alignment Module (*i.e.*, $\mathcal{L}_{ras}$). The first module encourages global semantic consistency between the codebook and text. The second module aims to transfer the rich semantic relationship between words into codes. Next, we introduce these two modules in detail.

### 3.2.1 Semantic Alignment Module

Considering that paired image and text data have consistent semantic information and the missing information of masked data can be completed from the other modality, we propose global semantic alignment, which aims to enhance the consistency of global semantics between text and visual codes, and masked text prediction, which uses visual codes to restore the masked words. Next, we discuss how to align text and codebook in the semantic space.

**Text Information Encoder**: Instead of jointly training text and codebook from scratch, we employ a pre-trained cross-modal model CLIP [32] to encode text information. Its advantage is that such text information already has good cross-modal semantic knowledge and is beneficial for codebook learning. Specifically, for a given text description of an image $t = \{w_{SOT}, w_1, w_2, \cdots, w_{n-2}, w_{EOT}\}$, where $w_i$ denotes the $i$-th word, $w_{SOT}$ and $w_{EOT}$ represent the $[start]$ token and $[end]$ token, respectively, and $n$ is text sequence length. We use the text encoder of a pre-trained CLIP model to obtain whole sequence embedding $T \in \mathbb{R}^{n \times d_t}$:

$$T = \{e_{SOT}, e_{w_1}, e_{w_2}, \cdots, e_{w_n}, e_{EOT}\} = \text{CLIP}(t). \tag{3}$$

Similar to CLIP, we use the $e_{EOT}$ to represent the global context feature of the sequence.

**Global Semantic Alignment** aims to align text and image visual codes in the global semantic space. For getting the global representation of visual codes, we employ a vision transformer (ViT) $f_{\theta_{vt}}$ [8] to encode the discrete codes of image. Specifically, given an image, we firstly obtain the discrete codes of image $Z$ by Eq. 1. Then, we introduce a learnable global token $[CLS]$ at the beginning to form a token sequence $Z_c$, where global token $[CLS]$ is employed to capture the image's global context information. We feed the sequence into $f_{\theta_{vt}}$ to get a new visual code representation, that is:

$$Z_{vt} = \{e_{CLS}, e_1, e_2, \cdots, e_{\frac{H}{f} \times \frac{W}{f}}\} = f_{\theta_{vt}}(Z_c). \tag{4}$$

Finally, we employ InfoNCE [29], which maximizes the similarity between visual and text in the global representation, as our learning objective, where $\mathcal{B}$ is the batch size, $s(\cdot, \cdot)$ is cosine similarity:

$$\mathcal{L}_{gsa} = -\sum_{i \in \mathcal{B}} \log \frac{\exp(s(e_{CLS}^i, e_{EOT}^i))}{\sum_{j \in \mathcal{B}} \exp(s(e_{CLS}^i, e_{EOT}^j))}. \tag{5}$$

**Masked Text Prediction**: To further enhance the semantic alignment, we propose to use discrete visual codes to reconstruct the masked words from a more fine-grained perspective, refer to Fig. 2 left. Formally, for a given fixed-length text sequence of $n-2$, we first randomly sample the masking ratio $r$ from a truncated Gaussian distribution [19]. Subsequently, we randomly mask out $r \cdot (n-2)$ words and replace them with learnable $[mask_i]$ tokens based on their positions $i$. Next, a self-attention module [44] is employed to learn adaptive masked word embeddings based on unmasked words. The resulting adaptive masked sequence is denoted as $T^{msk} = \{e_{SOT}, m_1, e_{w_2}, m_3, \cdots, e_{EOT}\}$, where $m_i$ is the mask token embedding at the $i$-th position in the sequence. Following this, a cross attention decoder $f_{\theta_m}(\cdot, \cdot)$ is employed to predict the masked word tokens given the discrete visual codes $Z_{vt}$ obtained by Eq. 4. Finally, we add a cross-entropy loss $H(\cdot, \cdot)$ between the ground-truth word tokens and the output of the decoder. Let $y_{msk}$ denote a one-hot vocabulary distribution where the ground-truth word token has a probability of 1, $f_{\theta_m}(Z_{vt}, T^{msk})$ denote the predicted probability of model for masked word tokens. That is:

$$\mathcal{L}_{mtp} = -\mathbb{E}_{(Z_{vt}, T^{msk}) \sim \mathcal{B}} H(y_{msk}, f_{\theta_m}(Z_{vt}, T^{msk})). \tag{6}$$

### 3.2.2 Relationship Alignment Module

While the two aforementioned loss functions for achieving good alignment at holistic semantic space have demonstrated initial promise, they cannot satisfy more complex reasoning tasks like image captioning and VQA. Inspired by some VQA techniques [26, 40, 24, 1], the semantic relationships between pre-trained words play a very important role in complex text reasoning tasks. For instance, as shown in Fig 1, to answer question ("What is this woman holding?"), one needs to fully understand the visual objects "women", "racket", and semantic relationship between them ("holding"). Based on the above fact, we propose to transfer the semantic relationship between words into codes. Such semantic relationships enable the model to better understand the image for addressing complex reasoning tasks.

But unfortunately, there is an issue there is no alignment between words and codes. Thanks for the above two losses that have provided semantic alignment of text and visual codes. To achieve the above idea, as shown in Fig. 3, we first use $Z_{vt}$ to align with words. Then, we inject semantic relationships between words into the initial codebook $Z$, instead of the $Z_{vt}$. Its advantage is it can prevent codes from collapsing into a single point for learning more diverse representations by relationship limiting. Then, $Z_{vt}$ primarily serves the purpose of aligning words and codes, but it is a crucial step for subsequent processes. Specifically, given any two words of a sentence, we use pre-trained word embedding [32] to encode words, $e_{w_i}$ and $e_{w_j}$. We employ cosine similarity to find the index of the code from $Z_{vt}$ that is most similar to the word. Then, one can get code embedding from $Z$ based on the index:

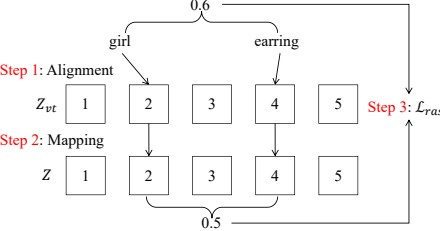

Figure 3: Illustration of relationship alignment module, we use $Z_{vt}$ to align with two words, then inject the semantic relationship of two words into $Z$ codes.

$$e_{z_i} = Z[\underset{e_z \in Z_{vt[1:]}}{argmax} \, s(e_{w_i}, e_z), :], \qquad e_{z_j} = Z[\underset{e_z \in Z_{vt[1:]}}{argmax} \, s(e_{w_j}, e_z), :]. \tag{7}$$

Next, we consider cosine similarity as a measure of semantic relationships between words and leverage it to establish corresponding relationships between codes achieving semantic relationship transfer. Finally, we utilize the following loss function as learning objective:

$$\mathcal{L}_{ras} = \sum_{(w_i, w_j) \in t} \left( s(e_{w_i}, e_{w_j}) - s(e_{z_i}, e_{z_j}) \right)^2. \tag{8}$$

### 3.2.3 Training Objective

We use three hyperparameters (*i.e.*, $\alpha$, $\beta$, and $\gamma$) to control three losses, respectively. Finally, the overall objective function is:

$$\mathcal{L} = \mathcal{L}_{vq} + \alpha \mathcal{L}_{gsa} + \beta \mathcal{L}_{mtp} + \gamma \mathcal{L}_{ras}. \tag{9}$$

Table 1: Results of image reconstruction on TextCaps, CelebA-HQ, CUB-200, and MS-COCO. "VQ-VAE+LG" denotes considering our method LG-VQ based on VQ-VAE.

| Models | TextCaps | | CelebA-HQ | | CUB-200 | | MS-COCO | |
|---|---|---|---|---|---|---|---|---|
| | FID↓ | PSNR↑ | FID↓ | PSNR↑ | FID↓ | PSNR↑ | FID↓ | PSNR↑ |
| VQ-VAE | 82.31 | **21.96** | 41.45 | **25.57** | 54.92 | 24.38 | 86.21 | **23.55** |
| VQ-VAE+LG | **81.93** | 21.95 | **40.53** | 25.04 | **36.55** | 25.60 | **79.54** | 23.40 |
| VQ-GAN | 24.08 | 19.64 | 5.66 | **24.10** | 3.63 | 22.19 | 14.45 | 20.21 |
| VQ-GAN+LG | **20.35** | **19.92** | **5.34** | 23.75 | **3.08** | **22.47** | **10.72** | **20.50** |
| CVQ | 16.35 | **20.24** | 5.19 | 23.15 | 3.61 | 22.29 | 9.94 | 20.48 |
| CVQ+LG | **15.51** | 20.21 | **4.90** | **24.48** | **3.33** | **22.47** | **9.69** | **20.71** |

# 4 Experiments

## 4.1 Experimental Settings

**Evaluation Metrics**. As our method is model-agnostic, we choose recent models, including VQ-VAE [43], VQ-GAN [9], and CVQ [53] as our backbone network. Following existing works [52, 53], we evaluate the reconstruction image quality on two evaluation metrics, *i.e.*, Fréchet Inception Distance (FID) [13] which evaluates the perceptual similarity of reconstructed images and original images, and Peak Signal-to-noise Ratio (PSNR) [10] is employed to measure the pixel-level similarity between the reconstructed and original images.

**Dataset**. We evaluate our method on four public datasets, including TextCaps [39], CelebA-HQ [23], CUB-200 [45], and MS-COCO [20]. For CelebA-HQ, CUB-200, and MS-COCO datasets, we use publicly available image captions, CelebA-HQ from [47], CUB-200 from [34], MS-COCO from [3]

**Implementation Details**. Following VQ-GAN [9], all images are reshaped $256 \times 256$ for reconstruction and generation. Down-sampling factor $f$ is set to 16. The codebook size $K$ is 1024. The batch size is 8. In our experiments, we maintain consistent parameter settings between our method LG-VQ and the chosen backbone networks (*i.e.*, VQ-VAE [43], VQ-GAN [9], and CVQ [53]) for a fair comparison. For each image, we randomly select a text from multi-text for training. Since our method introduces additional text and pre-trained CLIP model, for a fair comparison, we select VQCT [51] as the baseline for various downstream tasks. VQCT extracts many visual-related words from a large amount of text and designs a novel codebook transfer network based on the pre-trained CLIP model to learn the visual codebook.

Table 2: Ablation study of our three loss functions on TextCaps and CUB-200.

| | Setting | TextCaps | CUB-200 |
|---|---|---|---|
| | | FID↓ | FID↓ |
| (i) | Baseline(VQ-GAN) | 24.08 | 3.63 |
| (ii) | $+ \mathcal{L}_{gsa}$ | 23.01 | 3.39 |
| (iii) | $+ \mathcal{L}_{mtp}$ | 21.54 | 3.49 |
| (iv) | $+ \mathcal{L}_{mtp} + \mathcal{L}_{ras}$ | 20.77 | 3.32 |
| (v) | $+ \mathcal{L}_{mtp} + \mathcal{L}_{gsa}$ | 20.46 | 3.34 |
| (vi) | $+ \mathcal{L}_{mtp} + \mathcal{L}_{gsa} + \mathcal{L}_{ras}$ | 20.35 | 3.08 |

Table 3: Results (Recall@1) of masked word prediction on CelebA-HQ and CUB-200. "Mask-1" denotes that text is randomly masked one word.

| Dataset | | Recall@1 |
|---|---|---|
| CelebA-HQ | Mask-1 | 99.55 |
| | Mask-3 | 99.24 |
| CUB-200 | Mask-1 | 83.65 |
| | Mask-3 | 80.17 |

## 4.2 Discussion of Results

Table 1 illustrates the image reconstruction performance of our model compared to the backbone model on multiple datasets. It can be observed that our method LG-VQ outperforms all compared methods on most evaluations, which suggests that our method is extremely effective and has strong generality. Compared with FID, our PSNR improvement is marginal, this is reasonable in the VQ research domain, which widely exists in previous VQ methods [21, 52, 53]. The main reason is that PSNR only measures the pixel-level similarity of the images, while FID can effectively measure the diversity and semantic similarity of image generation. Compared with backbone models, the key difference lies in that our method introduces well pre-trained text semantics, which is beneficial to learning a more expressive codebook. This shows the effectiveness of our method. We also provide a qualitative comparison of the image reconstruction performance of different methods, please refer to

## 4.3 Ablation Study

**Are our three loss functions both effective?** In Table 2, we conduct an ablation study to show the effectiveness of the proposed three loss functions. Specifically, the VQ-GAN serves as the baseline model (*i.e.*, without introducing any loss). We do not conduct a separate experiment on $\mathcal{L}_{ras}$ because this module requires code and words to be well aligned. Based on the results from (i) $\sim$ (vi), we draw several key conclusions: Firstly, each loss function plays a crucial role in improving the performance of image reconstruction. Secondly, the performance of (iii) outperforms (ii) by a large margin on TextCaps. This is reasonable because TextCaps's texts are richer and more diverse than CUB-200, it can provide more knowledge for more fine-grained alignment between codes and text, which is useful for the learning of a more robust codebook. Thirdly, analyzing the results of (iii) and (iv), injecting word-level semantic relationships into codes is beneficial, which confirms our motivation. Furthermore, the performance of (v) outperforms (i), which is reasonable because the abundant semantic knowledge from pre-trained text can be fully exploited for learning more robust codebook representation. This supports the motivation of learning a multi-modal codebook (*i.e.*, aligned with text). Finally, comparing the results of (vi) with (i)$\sim$(v), fully considering all losses achieves the best performance, indicating the effectiveness of our method.

**Can our global semantic supervision align vision and language?** In Fig. 4, we provide several image-to-text retrieval cases on CelebA-HQ and CUB-200 datasets based on VQ-GAN+LG. From the figure, it can be observed that our method can accurately retrieve text very similar to the image content, achieving the alignment of vision and language. For example, row 2 examples show that our method can precisely understand some key attributes of images (*e.g.*, "gray hair", "necktie", "big nose" and "chubby") and retrieve similar text. This suggests that the codes learned through our method have obtained good alignment with the text, which verifies the effectiveness of our method. Moreover, such alignment is beneficial for learning robust code representations and enhancing performance in multi-modal downstream tasks.

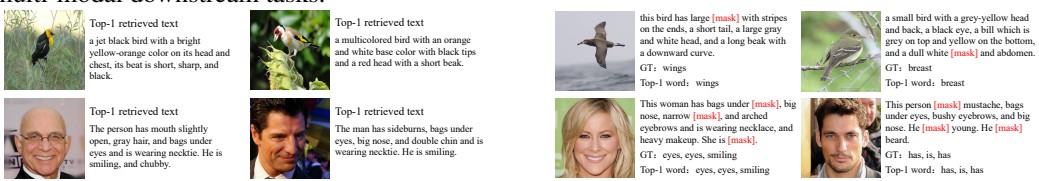

Figure 4: Examples of the top-1 most similar text selected on image-to-text retrieval task.

Figure 5: Examples of the top-1 word predicted on masked word prediction task.

**Can our codebook accurately predict masked words?** To answer this question, we conduct a word prediction task on test data based on VQ-GAN+LG by randomly masking one word or three words of text, as shown in Table 3. We use Recall@1 as the evaluation metric [18]. From the table, our method demonstrates accurate predictions of masked words, confirming the effectiveness of our approach. Fine-grained word prediction can help the codebook better understand the text semantics, which is crucial for improving the performance of the downstream task. Additionally, several examples in Fig. 5 demonstrate our method's ability that accurately predict subject words (*e.g.*, wings, eyes) and verbs (*e.g.*, has, is, and smiling), further affirming its strong multi-modal understanding capabilities.

**Can our codebook learn the word semantic relationships?**
In Fig. 6, we visualize the cosine similarity between words and the cosine similarity between codes aligned with the words for a certain sample based on VQ-GAN+LG. From the figure, we can see our codes can learn consistent relationships with word semantics compared with VQ-GAN. For example, the similarity "code 33" vs "code 232" (0.46) resembles "wings" vs "chest" (0.49). In addition, we provide a quantitative similarity evaluation between codes and words in Table 4. From the results, we can find that our codes indeed achieve consistent semantic relationships with words.

Table 4: Results of similarity evaluation between codes and words on CUB-200 all test data.

| Method | VQ-GAN | VQ-GAN+LG |
|--------|--------|-----------|
| MSE↓   | 0.6374 | 0.0351    |

**Is our method effectively learning more diverse code representation?** Following [52], we directly feed each codebook embedding $e_k$ (size: $1 \times 1 \times 256$) into the decoder $D_{\theta_d}(\cdot)$ to generate codebook

image (size: $16 \times 16 \times 3$). Then, we concatenate all codebook images to form a big image with $32 \times 32$ patches. Finally, we visualize the result of VQ-GAN and our LG-VQ on TextCaps and MS-COCO as shown in Fig. 7. This visualization suggests that our method enables the model to learn more diverse code representations and improve codebook usage.

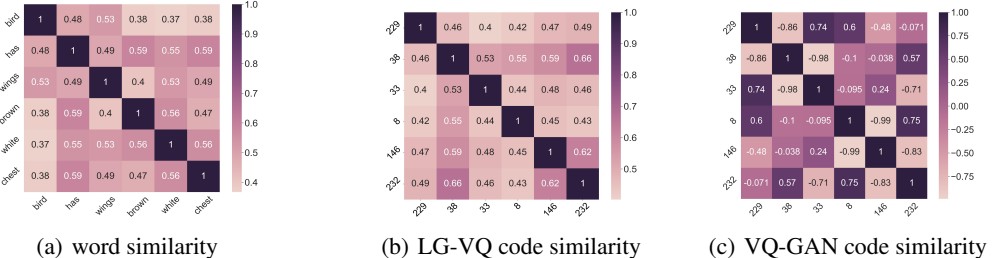

| (a) word similarity | (b) LG-VQ code similarity | (c) VQ-GAN code similarity |

Figure 6: Visualization of words similarity and image codes similarity aligned with the word. We extract some representative words from the text as a demonstration.

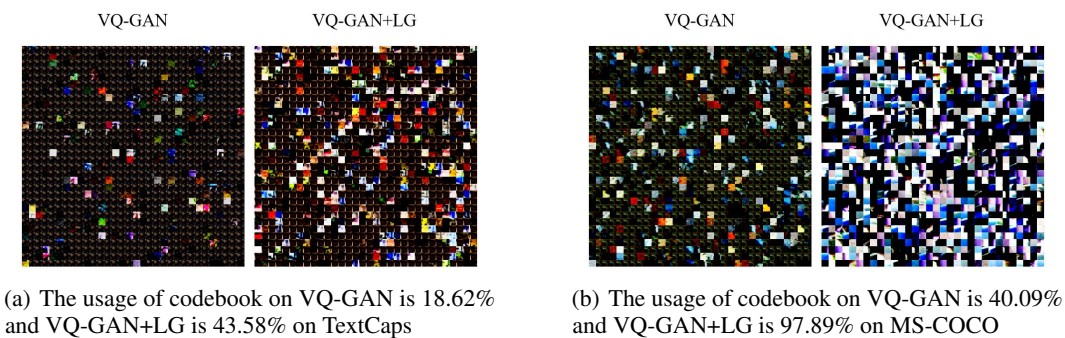

(a) The usage of codebook on VQ-GAN is 18.62% and VQ-GAN+LG is 43.58% on TextCaps

(b) The usage of codebook on VQ-GAN is 40.09% and VQ-GAN+LG is 97.89% on MS-COCO

Figure 7: Visualization of the codebook of VQ-GAN and LG-VQ on TextCaps and MS-COCO.

### 4.4 Application

#### 4.4.1 Image Generation

Following [9, 37, 11], we conduct image generation downstream tasks (*i.e.*, text-to-image, semantic synthesis, unconditional generation, and image completion) to fully validate the effectiveness of the learned codebook on CelebA-HQ.

**Text-to-Image**. In Table 5, we compare our LG-VQ with the state-of-the-art models on CelebA-HQ dataset for text-to-image. From the results, our LG-VQ method outperforms baseline methods by a large margin. This is reasonable due to the incorporation of pre-trained text knowledge enabling a comprehensive understanding of the text, which suggests our method's effectiveness. Moreover, we provide some synthesis examples comparing the results of our LG-VQ with baseline methods in Figure 8, showing the performance in the text-to-image task. From the figure, we can see our method not only comprehensively understands the given text conditions but also excels in generating realistic images compared with baseline methods. For instance, our method can capture the "glasses", "man", "long black hair", and "no beard" key attributions.

**Semantic Synthesis**. Following [9], we compare with existing semantic synthesis models in Table 6. Our method achieves the best performance, which suggests our method's effectiveness. We provide some examples in Appendix Figure 13.

**Unconditional Generation and Image Completion**. Following [9], we conduct unconditional image generation and Image Completion on CelebA-HQ dataset, as shown in Table 7 and Table 10. From the results, we can see that our method can significantly improve the performance of VQ-GAN, which is reasonable because pre-trained text can provide rich semantic knowledge for learning more robust codebook representation. This suggests the effectiveness of our method. We provide some examples in Appendix Figure 18 and Figure 17.

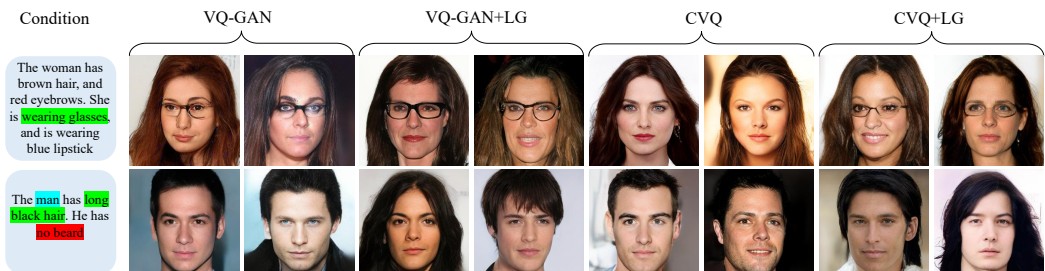

Figure 8: Text-to-image synthesis and semantic image synthesis on CelebA-HQ. Text with background color emphasizes generated details

### 4.4.2 Visual Text Reasoning

Follow [27, 6], we use the learned codebook to conduct two visual text reasoning tasks: 1) image captioning on CUB-200; and 2) visual question answering (VQA) on COCO-QA [35]. For the experimental setting, please refer to Appendix A.1.

**Image captioning**. Following [27], we conduct the image captioning task on the CUB-200 dataset. We compare two recent work V2L Tokenizer [56] and VQCT [51]. We select VQ-GAN as our backbone network. The results are shown in Table 11. For the results, we can see that our LG-VQ method outperforms the performance of VQ-GAN. This is reasonable because the pre-trained text provides rich context and relationship semantics for codebook learning, which verifies our motivation for learning a text-aligned codebook to improve the performance of the codebook on cross-modal tasks. On the other hand, the V2L Tokenizer and VQCT cannot achieve very good performance because it is difficult to assign correct semantic language tokens to images. Compared with the V2L Tokenizer, our method utilizes pre-trained text semantics as supervised information. Its advantage is can make the codebook learn semantic information consistent with the text (*i.e.*, learning a text-aligned codebook). And, our method is model-agnostic, which can be easily integrated into existing VQ models.

**Visual Question Answering**. We select VQ-GAN and VQCT [51] as the baseline. We conduct the VQA task on the COCO-QA [35] using the codebook trained on the MS-COCO dataset. The results are shown in Table 9. From the results, we can see that our LG-VQ method significantly improves the performance of VQ-GAN on VQA task (approximately 8.32%↑ on Accuracy). That is reasonable due to we introduce pre-trained text semantics to enable us to obtain a codebook aligned with the text, which is helpful for comprehensively understanding the given text question. This confirms our motivation and the effectiveness of our method.

Table 5: Results of text-to-image on CelebA-HQ.

| Model | Text-to-Image FID↓ |
|---|---|
| Unite and Conqu [4] | 26.09 |
| Corgi [54] | 19.74 |
| LAFITE [55] | 12.54 |
| VQ-GAN | 15.29 |
| CVQ | 13.23 |
| VQ-GAN+LG | 12.61 |
| CVQ+LG | 12.33 |

Table 6: Result (FID↓) of semantic synthesis on CelebA-HQ.

| Model | Semantic Synthesis FID↓ |
|---|---|
| Reg-VQ [52] | 15.34 |
| VQCT [51] | 14.47 |
| VQ-GAN | 11.53 |
| CVQ | 11.04 |
| VQ-GAN+LG | 11.46 |
| CVQ+LG | 11.03 |

### 4.4.3 Visual Grounding

We conduct a visual grounding task on refcoco dataset [49] to validate the effectiveness of the learned MS-COCO's codebook. Following the same metric used in [5], a prediction is right if the IoU between the grounding-truth box and the predicted bounding box is larger than 0.5. We select VQ-GAN and VQCT [51] as the baseline. The results are shown in Table 8. From the results, we can see that the performance of our method consistently outperforms VQ-GAN and VQCT, which suggests its effectiveness. We also provide a qualitative comparison in Appendix Figure 19. For the experimental setting, please refer to Appendix A.1.

Table 7: Result (FID↓) of un-conditional image generation on CelebA-HQ.

| Model | CelebA-HQ FID↓ |
|---|---|
| Style ALAE [30] | 19.2 |
| DC-VAE [31] | 15.8 |
| VQ-GAN | 10.2 |
| LG-VQ | 9.1 |

Table 8: Result (FID↓) of visual grounding on refcoco dataset using MS-COCO's codebook.

| Model | Visual Grounding Accuracy(0.5)↑ |
|---|---|
| VQ-GAN | 9.14 |
| VQCT [51] | 9.46 |
| LG-VQ | 9.62 |

Table 9: Results of (Accuracy and WUPS [46]) VQA on COCO-QA [35] dataset using MS-COCO's codebook.

| Setting | VQA | |
|---|---|---|
| | Accuracy↑ | WUPS↑ |
| VQCT [51] | 40.42 | 82.06 |
| VQ-GAN | 37.82 | 83.22 |
| LG-VQ | 40.97 | 83.56 |

Table 10: Result (FID↓) of image completion on CelebA-HQ.

| Model | CelebA-HQ FID↓ |
|---|---|
| VQ-GAN | 9.02 |
| LG-VQ | 8.14 |
| Improve | **9.76%** |

Table 11: Results of image captioning on CUB-200.

| Model | Image Captioning | | | |
|---|---|---|---|---|
| | BLEU4↑ | ROUGE-L↑ | METEOR↑ | CIDEr-D↑ |
| VQ-GAN | 1.29 | 33.40 | 24.47 | 93.62 |
| V2L Tokenizer [56] | 1.59 | 30.65 | 25.76 | **104.14** |
| VQCT [51] | 1.38 | 26.50 | 24.63 | 98.22 |
| LG-VQ | **1.69** | **34.73** | **25.78** | 102.77 |

## 5 Conclusions

In this paper, we propose a novel codebook learning method, named LG-VQ. LG-VQ is a model-agnostic method and can easily be integrated into existing VQ models. In particular, we propose to incorporate pre-trained text semantics into the codebook by two novel supervision modules, *i.e.*, semantic and relationship. Quantitative and qualitative experiments demonstrate the strong generality of our method, showing its ability to improve the performance of the codebook in cross-modal tasks. **Limitations**. In our current paper, we suppose each word aligns with a code, but it fails to capture some more complex relationships between words and codes (*e.g.*, one code aligns with multiple words). In the future, we plan to investigate the relationships between codes and words. Moreover, although our results show that the performance of VQ in visual text reasoning tasks can be significantly improved, its results are still far lower than the performance of image captioning or VQA models. **Broader impact** Our paper shows that learning a multi-modal codebook (*i.e.*, a text-aligned codebook) can not only significantly improve the performance of reconstruction but also the performance of the codebook on cross-modal tasks. The potential impact of our research lies in its influence on future studies, specifically in the area of unified modeling of multi-modal understanding and generation. For instance, our work can be extended to interact with LLMs to improve multi-modal understanding and generation capabilities. In particular, our model can be used to generate images or text. It may be exploited to produce some erroneous and unethical information, which needs to be handled carefully before employing our model in practical applications.

## Acknowledgement

This work was supported by the Shenzhen Peacock Program under Grant No. ZX20230597, NSFC under Grant No. 62272130 and Grant No. 62376072, and the Shenzhen Science and Technology Program under Grant No. KCXFZ20211020163403005. It was also supported by the Major Key Project of PCL (PCL2023A08) and the National Science Foundation of China: Multi-source Cross-platform Video Analysis and Understanding for Intelligent Perception in Smart City: U20B2052.

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

# A    Appendix / supplemental material

## A.1    Experiment Details

**Semantic image synthesis, unconditional generation and image completion**: All models follow the default setting of VQ-GAN-Transformer[2]. Specifically, the vocabulary size, embedding number, and input sequence length are 1024, 1024, and 512, respectively. The layers and heads of the transformer are both 16. The semantic image synthesis experiments are conducted on 1 4090 GPU with a batch size of 15 and one day of training time. The unconditional generation and image completion experiments are conducted on 2 4090 GPUs with a batch size of 36 and one day of training time.

**Text-to-image generation**: All models follow the default setting of VQ-Diffusion[3]. Specifically, the layers of the transformer are 19 with dimension of 1024. The diffusion step is 100. The training epoch is 90 for all models. The experiments are conducted on 1 4090 GPU with a batch size of 24 and two days of training time.

**Image captioning**: Inspired by ClipCap [27], we use the trained codes to replace the ClipCap's prefix embeddings. The model framework is shown in Fig. 9 (a). The training epoch is 100 for all models. The experiments are conducted on 2 4090 GPUs with a batch size of 60 and one day of training time.

**Visual question answering**: The COCO-QA [35] dataset is automatically generated from captions in the Microsoft COCO dataset [20]. There are 78,736 train questions and 38,948 test questions in the dataset, These questions are based on 8,000 and 4,000 images respectively. There are four types of questions including object, number, color, and location. Each type takes 70%, 7%, 17%, and 6% of the whole dataset, respectively. All answers in this data set are single word. Following the image captioning task, we use the last hidden embedding to do VQA, as shown in Fig. 9 (b). Following the [24], we report classification accuracy and Wu-Palmer similarity (WUPS). The training epoch is 50 for all models. The experiments are conducted on 2 4090 GPUs with a batch size of 60 and one day of training time.

**Visual Grounding**: The refcoco dataset [49] includes 19,994 images with 50,000 referred objects. Each object has more than one referring expression, and there are 142,210 referring expressions in this dataset. There are two commonly used split protocols for this dataset. One is RefCOCOg-google [25], and the other is RefCOCOgumd [28]. We follow RefCOCOgumd [28] to split the dataset. The train set has 42,404 expressions, the validation set has 3,811 expressions, and the test set has 3,785 expressions. Following [5], we concatenate the image codes and text tokens and feed them into a learnable transformer with coordinate regression layers (*i.e.*, FNN) to predict the object box. The training epoch is 100 for all models. The experiments are conducted on 2 4090 GPUs with a batch size of 30 and several hours of training time.

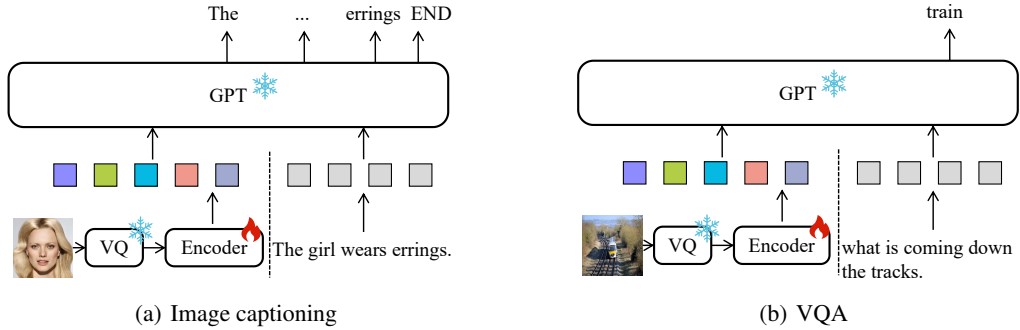

(a) Image captioning                    (b) VQA

Figure 9: The architecture of the visual text reasoning based on GPT.

---

[2]https://github.com/CompVis/taming-transformers
[3]https://github.com/microsoft/VQ-Diffusion

## A.2 Experimental comparison with VQCT

We provide comparisons with VQCT [51] for image reconstruction in Table 12. Moreover, we also provide experimental results on further integrating our method into VQCT to verify its effectiveness and versatility. The results are shown in Table 13.

Table 12: Comparison of VQCT [51] and our method on reconstruction

| Model | Codebook Size | #Tokens | CelebA-HQ | CUB-200 | MS-COCO |
|---|---|---|---|---|---|
| VQCT | 6207 | 512 | 5.02 | **2.13** | 9.82 |
| VQ-GAN+LG | 1024 | 256 | 5.34 | 3.08 | 10.72 |
| CVQ+LG | 1024 | 256 | **4.90** | 3.33 | **9.69** |

Table 13: Comparison of **reconstruction and VQA** on VQCT and VQCT+LG on the MS-COCO dataset.

| | Image Reconstruction | VQA |
|---|---|---|
| Model | FID↓ | Accuracy↑ |
| VQCT | 9.82 | 40.42 |
| VQCT+LG | **9.57** | **40.64** |

## A.3 More Examples and Qualitative Results

We provide more examples of image reconstruction in Fig. 10, image-to-text retrieval in Fig. 11 and Fig. 12. We also provide more image synthesis results in Fig. 13 for semantic image synthesis, and text-to-image synthesis in Fig. 14. We provide some examples of image captioning in Fig. 15 and VQA in Fig. 16. We provide some examples of unconditional generation in Fig. 18, and image completion in Fig. 17. We also provide a qualitative comparison of visual grounding in Fig. 19.

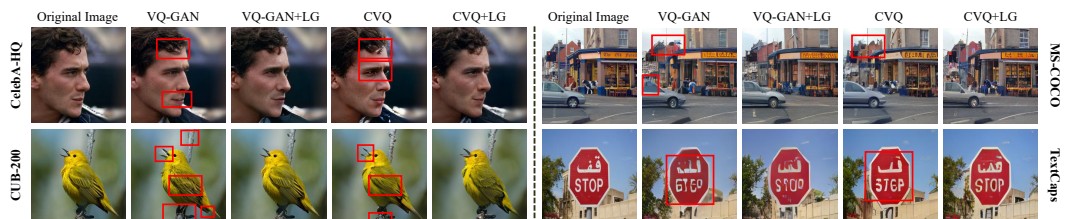

Figure 10: Reconstruction from different models on four datasets. The red-color boxes highlight reconstruction details.

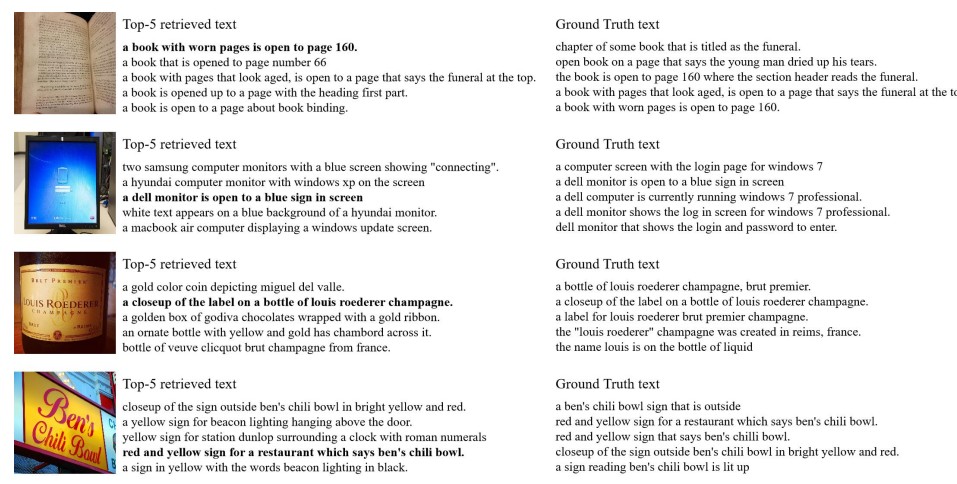

Figure 11: Examples of the top-5 most similar text selected on Textcaps based on VQ-GAN+LG. The bold text means the same as the ground truth result.

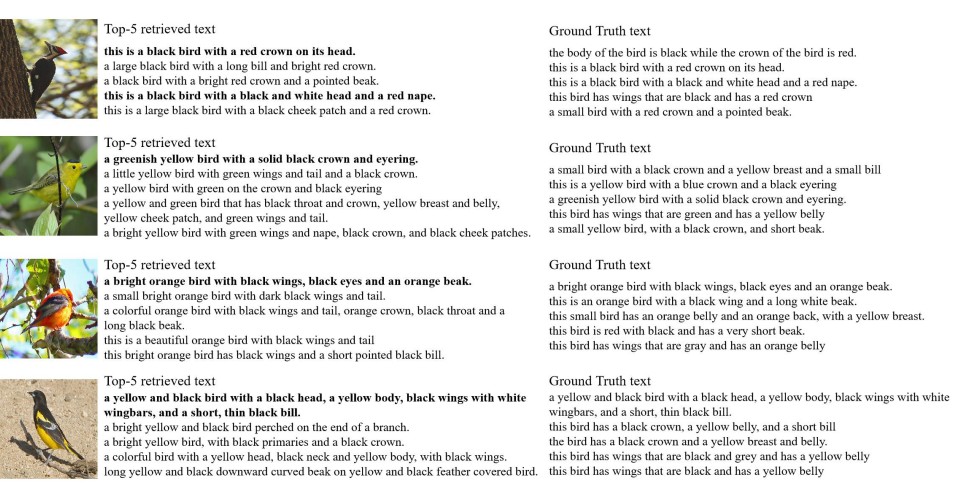

Figure 12: Examples of the top-5 most similar text selected on CUB-200 based on VQ-GAN+LG. The bold text means the same as the ground truth result.

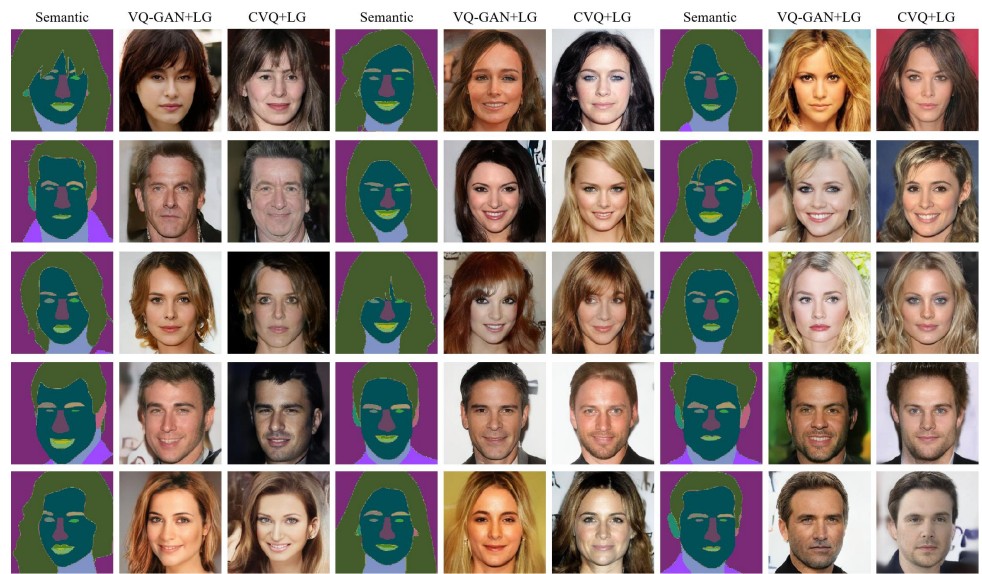

Figure 13: Semantic image synthesis on CelebA-HQ.

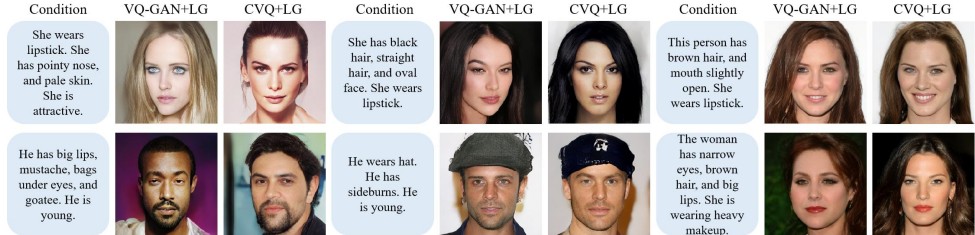

Figure 14: More text-to-image generation on CelebA-HQ.

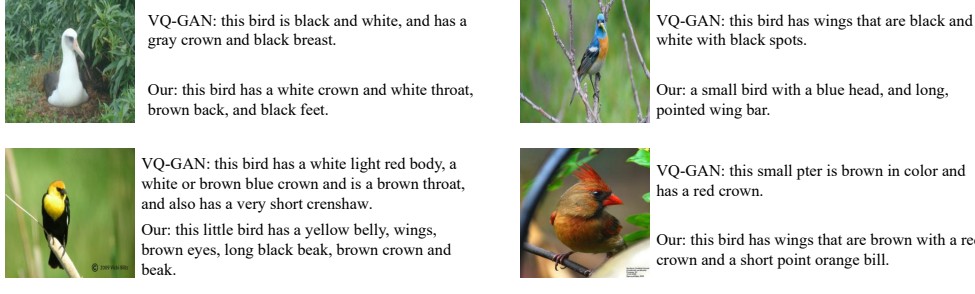

Figure 15: Image Captioning on CUB-200 based on VQ-GAN and VQ-GAN+LG.

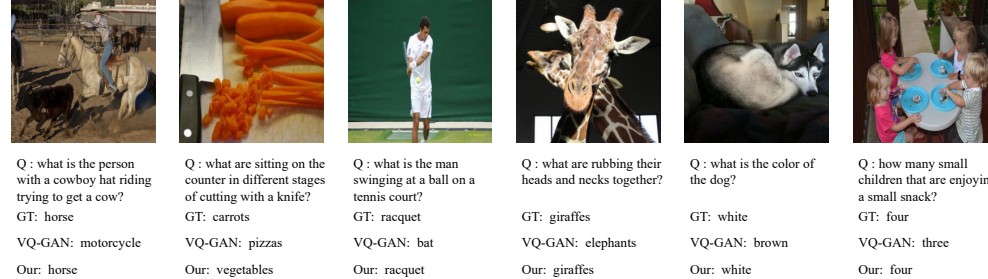

Figure 16: VQA on COCO-QA based on VQ-GAN and VQ-GAN+LG.

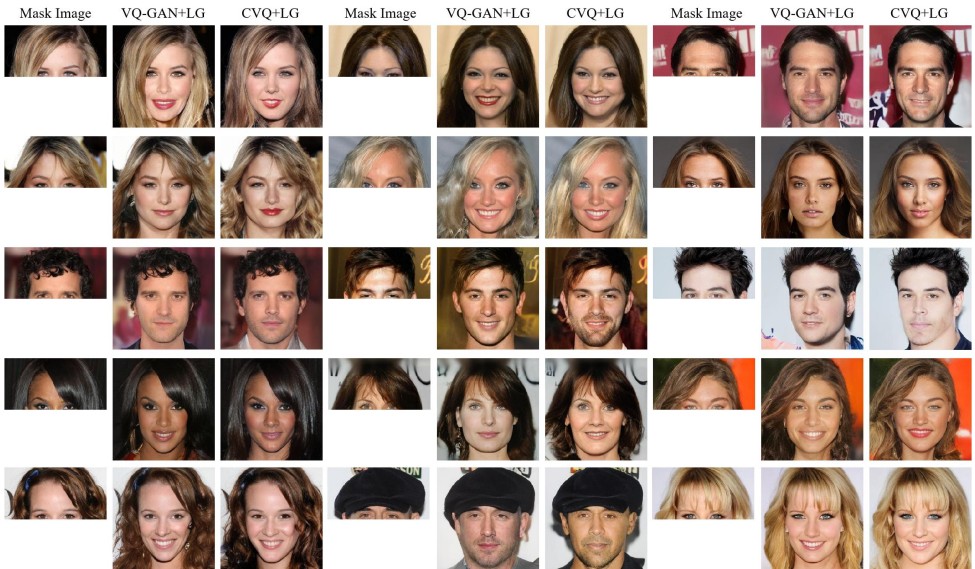

Figure 17: Image completion on CelebA-HQ.

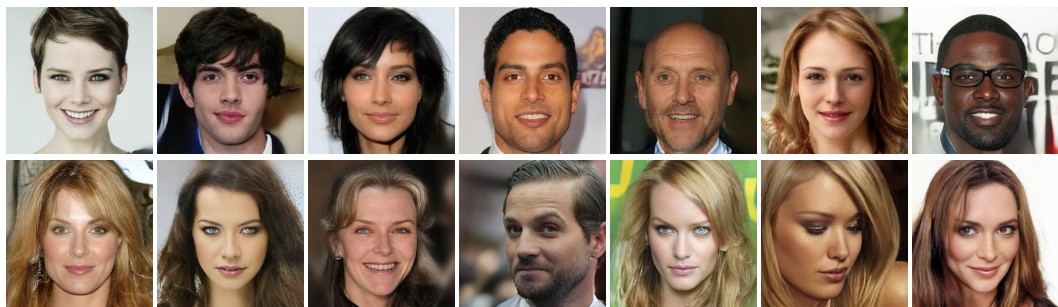

Figure 18: Examples of **unconditional image generation** on CelebA-HQ based on VQ-GAN+LG.

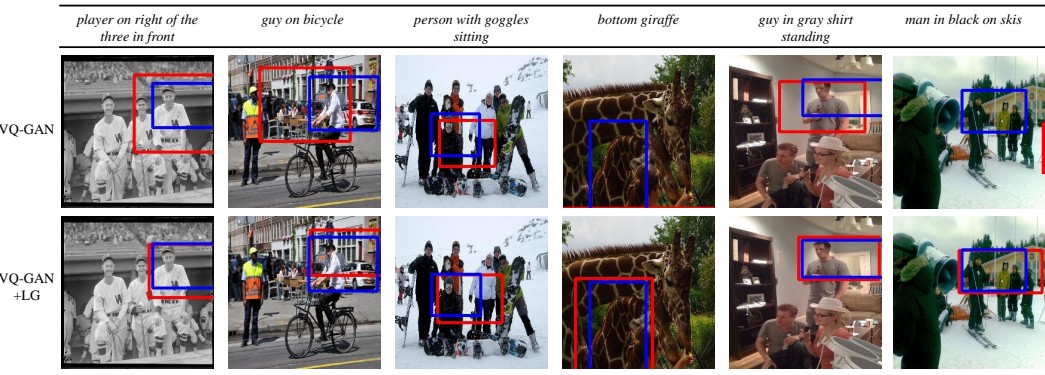

Figure 19: Examples of **visual grounding** on refcoco. Blue boxes are the ground-truth, red boxes are the model predictions.

