# OpenReview forum: "LG-VQ: Language-Guided Codebook Learning"
_NeurIPS.cc/2024/Conference — NeurIPS 2024 poster_

### Official Review · Reviewer_WVeZ · 2024-07-09

**Soundness:** 3
**Presentation:** 3
**Contribution:** 3
**Rating:** 6
**Confidence:** 4

**Summary:**

This paper points out that most of existing vector quantization methods learn a single-modal codebook, resulting in suboptimal performance when the codebook is applied to multi-modal downstream tasks. To address this issue, the authors propose a new language-guided codebook learning framework, called LG-VQ, which learns a codebook that can be aligned with the text to improve the performance of multimodal downstream tasks. Specifically, with the pretrained text semantics as prior knowledge, two novel alignment modules are devised to transfer such prior knowledge into codes for achieving codebook text alignment. The proposed method is evaluated on reconstruction and various multi-modal downstream tasks, and achieves superior performance.

**Strengths:**

1)	The authors point out the limitation of existing single-modal quantization methods and propose a novel multi-modal codebook learning method.
2)	Two semantic supervision modules are designed to effectively learn text-aligned codebook.
3)	Comprehensive experiments have been conducted and prove the effectiveness of the proposed method.
4)	The paper is well-written and easy to follow.

**Weaknesses:**

- The details of the relationship alignment module are not clear. Fig. 3 is not easy to understand. The sec. 3.2.2 should be reorganized to clearly introduce the relationship alignment module.
- There may exist errors in the denominator of Eq. (5).
- In the ablation study, the performance of only adopting the relationship alignment module (+L_{ras}) should also be evaluated. Could this module be used separately?

**Questions:**

please try to address the weaknesses.

**Limitations:**

n/a.

---

> ### Author Rebuttal · Authors · 2024-08-05
>
> We are thankful for the reviewer’s time and valuable feedback. We are glad that the reviewer found that our paper is easy to follow and well-written. Please see below for our responses to your comment and questions.
>
> > Q1: The details of the relationship alignment module are not clear. Fig. 3 is not easy to understand. The sec. 3.2.2 should be reorganized to clearly introduce the relationship alignment module.
>
>
> Thanks for your valuable comment. We provide a more detailed explanation of the relationship alignment module:
>
> **Motivation:**   Our motivation is that when the codebook is applied to the VQA task (e.g. to answer the question "What is this woman holding?" in Figure 1), the model needs to identify key visual objects "women" and "racket" from the image's codes, and also needs to understand the semantic relationship between the objects. However, existing VQ-based methods train codebook independently and do not align text, resulting in their poor performance.
>
> **Our idea:** The pre-trained word embeddings contain rich semantic relationship knowledge, which is beneficial to VQA tasks. Therefore, we propose a relationship alignment module that aims to transfer the semantic relationship between words into codes. which helps the model better understand images for complex reasoning tasks and improves alignment between codes and words.
>
> We provide a more detailed explanation of Figure 3:
>
> **Input**: given any two words of a sentence (e.g., 'girl' and 'earring'), we encode two words using pretrained CLIP model.
>
> **Alignment**:  we find two codes closest to two words, i.e., code 2 and code 4, based on the similarity between the word embeddings and $Z_{vq}$.
>
> **Mapping:**  we can get the code 2 embedding and code 4 embedding from Z based on index id (i.e., 2 and 4).
>
> **Relationship Transfer ($L_{ras}$):**  we constrain the cosine similarity of code 2 and code 4 embedding to match the cosine similarity between 'girl' and 'earring' embedding.
>
> We will reorganize Section 3.2.2, improve Figure 3,  and add it to our final version.
>
>
> > Q2: There may exist errors in the denominator of Eq. (5).
>
> Thanks for your construction comment. We will fix it in our final version.
>
>
>
> > Q3: In the ablation study, the performance of only adopting the relationship alignment module (+L_{ras}) should also be evaluated. Could this module be used separately?
>
> Following your suggestions, we conduction an ablation study on $L_{ras}$ on TextCaps, as shown in Table R4.1. Specifically, we use a layer of MLP network to replace $Z_{vq}$ to learn the alignment between code and word. This result suggests our $L_{ras}$ is effective.
>
> **Table R4.1 Ablation study of our $L_{ras}$ loss function on TextCaps.**
>
> |                                           |       TextCaps       |
> | ----------------------------------------- | :------------------: |
> | Setting                                   | FID$\downarrow$ |
> | VQ-GAN                                    |        24.08         |
> | + $\mathcal{L}_{ras}$                     |        23.81         |

---

### Official Review · Reviewer_YeAE · 2024-07-11

**Soundness:** 3
**Presentation:** 3
**Contribution:** 2
**Rating:** 5
**Confidence:** 2

**Summary:**

This paper considers the limitations in learning an expressive codebook and
proposes a novel multi-modal codebook learning method. The authors propose two semantic supervision modules and three losses. Experiments demonstrates the effectiveness of their method.

**Strengths:**

1.	This work is easy to follow, and its motivation is interesting.
2.	Experiments demonstrates the effectiveness of the method.

**Weaknesses:**

1. It is not easy to obtain major objects and relations between different objects. For example, in Figure 1, ‘the net’, the people beside the court, and the chairs are also very obvious objects. Is it necessary to extract the information of these objects as well?

2. The method shown in Figure 1 uses additional detectors to extract the visual features of the corresponding objects. However, there is no description of these detectors in the article, making it unclear.

3. The authors consider the relationship between text and feature embedding. The effectiveness of the method should be further validated on grounding tasks. (e.g., refcoco)

**Questions:**

1. How can the proposed method be used in downstream tasks, and will it incur additional computational overhead?

**Limitations:**

As stated by the authors, this work assumes each word aligns with a code, which offers a limited representation.

---

> ### Author Rebuttal · Authors · 2024-08-05
>
> We thank the reviewer for their time and valuable feedback. We are glad that the reviewer found that our paper is easy to follow and motivation is interesting. Please see below for our responses to your concerns and questions.
>
> > Q1: It is not easy to obtain major objects and relations between different objects. For example, in Figure 1, ‘the net’, the people beside the court, and the chairs are also very obvious objects. Is it necessary to extract the information of these objects as well?
>
> In fact, we assume that image captions contain the image's main objects, and the pretrained text semantics contain the relationship between objects. Moreover, we do not consider objects that exist in the image but not in the captions (e.g. 'the net' you mentioned), because it is not easy to obtain these objects as you mentioned.
>
>
>
> > Q2: The method shown in Figure 1 uses additional detectors to extract the visual features of the corresponding objects. However, there is no description of these detectors in the article, making it unclear.
>
> We would like to claim that we do not use any additional detectors. In our paper, we only introduce image captions. The main purpose of Figure 1 is to illustrate the motivation of our introduction of the relationship alignment module.
>
>
>
> >  Q3: The authors consider the relationship between text and feature embedding. The effectiveness of the method should be further validated on grounding tasks. (e.g., refcoco)
>
> Following your suggestions, we conduct visual grounding task on refcoco dataset using the codebook trained on the MS-COCO dataset. The results are shown in Table  R3.1 below, which confirm our method's effectiveness. We also provide a qualitative comparison in Figure C of [the rebuttal pdf](https://openreview.net/attachment?id=QfryItnYya&name=pdf). We will include these experimental results in our paper.
>
> **Table R3.1 Comparison of Visual Grounding on refcoco dataset.**
>
> |           | Visual Grounding on refcoco |
> | --------- | :-------------------------: |
> | Model     |   Accuracy(0.5)$\uparrow$    |
> | VQ-GAN    |            9.14             |
> | VQ-GAN+LG |            9.62             |
> | Improve   |          **5.25%**          |
>
> > Q4: How can the proposed method be used in downstream tasks, and will it incur additional computational overhead?
>
> We provide experiment details on how to leverage the learned codebooks for various downstream tasks in Appendix A.1 of our original paper. For all downstream tasks, we follow established experimental settings from existing works. For example, we follow VQ-GAN-Transformer for semantic image synthesis and VQ-diffusion for text-to-image.
>
> It does not incur any additional computational overhead when it is used for various downstream tasks. We provide the training parameters for reconstruction and downstream tasks as well as the approximate training time cost (seconds), as shown in Table R3.2 below. From the results, we can observe that our method only adds extra training parameters (i.e., 7.8M) and time (i.e., 0.04s) during reconstruction training, but both are negligible.
>
> **Table R3.2 Model training parameters and time cost in different tasks.**
>
> |           | Image Reconstruction |                       | Semantic Image Synthesis |                       |   Text-to-Image   |                       |
> | --------- | :------------------: | :-------------------: | :----------------------: | :-------------------: | :---------------: | :-------------------: |
> | Methods   |  #Train Params [M]   | Train time (sec/iter) |    #Train Params [M]     | Train time (sec/iter) | #Train Params [M] | Train time (sec/iter) |
> | VQ-GAN    |         74.9         |         0.83          |           204            |         0.46          |        496        |          0.6          |
> | VQ-GAN+LG |       74.9+7.8       |         0.87          |           204            |         0.46          |        496        |          0.6          |

---

> ### Comment · Reviewer_YeAE · 2024-08-14
>
> Thank you for your response. I have no other concerns with the paper itself, but since I’m not familiar with this domain, I will maintain my current rating and confidence.

---

> > ### Author Response · Authors · 2024-08-14
> > **Official Comment by Authors**
> >
> > We are thankful for the reviewer’s response and feedback.

---

### Official Review · Reviewer_heW4 · 2024-07-12

**Soundness:** 2
**Presentation:** 3
**Contribution:** 3
**Rating:** 5
**Confidence:** 3

**Summary:**

This paper proposed a new codebook learning method to improve the
performance of multi-modal downstream tasks.
The proposed method can be easily integrated into existing VQ models and shows performance improvements in various cross-modal tasks such as Image Generation and Visual Text Reasoning.

**Strengths:**

The paper is well-written and easy to follow.

The proposed method can be easily integrated into existing VQ models.

**Weaknesses:**

The main concern about this paper is its novelty. Although the method of enhancing performance by incorporating language-guided information has been proposed, there is concern that the performance improvement may come from the image-text knowledge of the pre-trained CLIP model rather than the proposed method itself.

The performance improvement is marginal.
LG-VQ method outperforms baseline methods in several tasks but performance improvement looks marginal.

**Questions:**

In the experiments section, there is only a performance comparison among VQ models such as VQ-VAE and VQ-GAN. A comparison with existing image generation models or visual text reasoning models is needed.

**Limitations:**

The authors discussed both the limitations and potential negative social impacts in the Conclusion section.

---

> ### Author Rebuttal · Authors · 2024-08-05
>
> We thank the reviewer for their time and valuable feedback. We are glad that the reviewer found that our paper is well-written and easy to follow. Please see below for our responses to your comments.
>
> > Q1: The main concern about this paper is its novelty.
>
> We would like to highlight the novelty of our paper from the following three aspects,
>
> - i. We **point out the limitations of existing VQ-based methods, which is that they learn a single-modal codebook**. This motivates us to propose a novel multi-modal codebook learning framework. This idea is very reasonable and novel.
> - ii. Our LG-VQ framework is designed **for the first time**, to utilize pretrained text semantics as supervised information to learn a text-aligned codebook. For the novelty, our whole framework design is not trivial and very challenging. Because of the need to fully understand the relationship between text and vision and the characteristics of various downstream tasks.   We thereby carefully design two alignment modules.  Moreover, these two modules are not independent of each other. The semantic alignment module serves the relationship alignment module. Finally, our method can be easily integrated into existing VQ-based models.
> - iii. From the experimental results, our method can **consistently improving performance** of existing VQ-based model codebook representation, reconstruction, and various downstream tasks.
>
> Based on the above three points, we believe that our work brings new insights for VQ-based model.
>
> > Q2: The performance improvement may come from the image-text knowledge of the pre-trained CLIP model rather than the proposed method itself.
>
> We select VQCT [1], a novel work recently accepted and published by CVPR-2024, as the baseline.  VQCT extracts many visual-related words from **a large amount of text**, and designs a novel codebook transfer network based on **the pretrained CLIP model** to learn the visual codebook. Detailed experimental results are provided in Table R2.1 below. The results suggest that the performance improvement comes from our method rather than only the text knowledge of the pre-trained CLIP model. We also provide experimental results on integrating our method into VQCT further to verify effectiveness and versatility of our method. Please refer to [the rebuttal pdf](https://openreview.net/attachment?id=QfryItnYya&name=pdf).
>
> **Table R2.1 Comparison of VQCT and our method on reconstruction, semantic synthesis, and visual text reasoning tasks.**
>
> | Model     | Codebook Size | #Tokens | CelebA-HQ | CUB-200  | MS-COCO  |
> | --------- | :-----------: | :-----: | :-------: | :------: | :------: |
> | VQCT      |     6207      |   512   |   5.02    | **2.13** |   9.82   |
> | VQ-GAN+LG |     1024      |   256   |   5.34    |   3.08   |  10.72   |
> | CVQ+LG    |     1024      |   256   | **4.90**  |   3.33   | **9.69** |
>
> |       | Semantic Synthesis on CelebA-HQ |                 | Image Captioning on CUB-200 |                  |                   |   VQA on MS-COCO   |
> | ----- | ------------------------------- | --------------- | --------------------------- | ---------------- | ----------------- | :----------------: |
> | Model | FID$\downarrow$                 | BLEU4$\uparrow$ | ROUGE-L$\uparrow$           | METEOR$\uparrow$ | CIDEr-D$\uparrow$ | Accuracy$\uparrow$ |
> | VQCT  | 14.47                           | 1.38            | 26.50                       | 24.63            | 98.22             |       40.42        |
> | Our   | **11.46**                       | **1.69**        | **34.73**                   | **25.78**        | **102.77**        |     **40.97**      |
>
>
>
>
>
> > Q3: The performance improvement is marginal.
>
> We would like to emphasize that the performance improvement of our method is not marginal. Here are the detailed performance gains of our VQ-GAN+LG compared to the baseline VQ-GAN:
>
> * Image Reconstruction (FID): Improvement of **1.87% to 15.49%**
>
> * Text-to-Image: Improvement of **17.52%**
>
> * Image Captioning: Improvement of **3.98% to 31.01%**
>
> * VQA (Accuracy): Improvement of **8.32%**
>
> - Image Completion: Improvement of **9.76%** (as detailed in our response to reviewer 8ZRY Q1 Table R1.2)
> - Unconditional Image Generation: Improvement of **10.78%** (as detailed in our response to reviewer 8ZRY Q1 Table R1.3)
>
>
>
> > Q4: Compared with other models in image generation and visual text reasoning.
>
> Following your suggestions, we provide a comparison with other models on the text-to-image (in Table R2.2) and VQA (in Table R2.3). From the results, we can see that our method achieves superior performance in image generation.  On VQA task, our method can significantly improve the performance of  VQ-GAN, which suggests our method is effective. But its results are still far lower than the performance of specialized VQA-based models. **This is shortcoming of the VQ-based model** and does not mean that our method is ineffective. We have acknowledged this limitation in Section 5 of our paper.
>
> **Table R2.2 Comparison of text-to-image.**
>
> |                     | Text-to-image on CelebA-HQ |
> | ------------------- | :------------------------: |
> | Model               |      FID$\downarrow$       |
> | Unite and Conqu [1] |           26.09            |
> | Corgi [2]           |           19.74            |
> | LAFITE [3]          |           12.54            |
> | VQ-GAN+LG           |           12.61            |
> | CVQ+LG              |         **12.33**          |
>
> **Table R2.3 Comparison of VQA.**
>
> |              |   VQA on COCO-QA   |
> | ------------ | :----------------: |
> | Model        | Accuracy$\uparrow$ |
> | HieCoAtt [4] |       62.50        |
> | OAM-VQA [5]  |       75.22        |
> | VQ-GAN       |       37.82        |
> | VQ-GAN+LG    |       40.97        |
>
>
>
> ------
>
> [1] Nair N G, et al. Unite and conquer. CVPR 2023.
>
> [2] Zhou Y, et al. Corgi. CVPR 2023.
>
> [3] Zhou Y, et al. LAFITE. CVPR 2022.
>
> [4] Lu J, et al. HieCoAtt. NIPS 2016.
>
> [5] Li P, et al. OAM-VQA. AAAI 2024.

---

> > ### Comment · Reviewer_heW4 · 2024-08-12
> >
> > Thank you to the authors for your detailed responses. I believe my concerns have been adequately addressed. I will proceed to revise the final score.

---

> > > ### Author Response · Authors · 2024-08-12
> > > **Official Comment by Authors**
> > >
> > > Thank you for your feedback and consideration!

---

### Official Review · Reviewer_8ZRy · 2024-07-13

**Soundness:** 3
**Presentation:** 3
**Contribution:** 3
**Rating:** 6
**Confidence:** 4

**Summary:**

This paper addresses the issue that current VQ codebook lacks multimodal information, and proposes to introduce language guidance to learn a more comprehensive one. It develops semantic alignment module and relationship alignment module with multiple additional losses to regularize the codebook with additional image captions and pretrained text encoders. The proposed method works on the codebook learning independently and can be applied to any VQ-based backbones. It is evaluated on various downstream applications and achieves enhanced performance compared to its baselines.

**Strengths:**

- The paper addresses the lack of multimodal information in current codebooks, which lacks of notice in previous work, and proposes novel modules to incorporate language information and enhances their semantic alignments.

- The proposed method is evaluated on various tasks and reaches good results surpassing baselines. Thorough ablation studies and analyses into its learned space are also performed.

**Weaknesses:**

- The proposed method involed additional language information (not only the pretrained CLIP model but also the raw language materials) during training, and in some downstream applications like the text-to-image generation, while most baselines/backbones don't incorporate such input at all. This might raise concerns on the comparison fairness and it's better to be specially noted.

- More non-language-conditioned image generation results should be displayed and compared with baselines/backbones, since the same reason above. According to the paper the language-guided codebook not only can parse additional text condition input but also itself has learned a better latent space overall, so this needs to be more clearly demonstrated. Figs. 13 and 17 show image-to-image translation without language, while they're not compared with any other methods. The most basic unconditional image generation should be evaluated and compared.

**Questions:**

- The proposed method requires the training images to have gt captions. How do you generate captions for image-only datasets?

- Is there a chance that the proposed method can be applied to other 2-stage generation frameworks, such as text-to-image latent diffusion's (especially the VQ-regularized) VAE? There is only language guidance on the diffusion process but not on the bottleneck latents/codebook.

**Limitations:**

- The limitations and border impacts are discussed in the paper Sec. 5.

---

> ### Author Rebuttal · Authors · 2024-07-31
>
> We thank Reviewer 8ZRy for the positive review and valuable feedback. We would like to address your questions below one by one.
>
> > Q1: The proposed method involved additional language information (not only the pretrained CLIP model but also the raw language materials) during training. This might raise concerns on the comparison fairness.
>
> Thank you again for taking the time to provide your valuable comments. For a fair comparison, we select VQCT[1], a novel work recently accepted and published by CVPR-2024, as the baseline. VQCT extracts many visual-related words from **a large amount of text**, and designs a novel codebook transfer network based on **the pretrained CLIP model** to learn the visual codebook. We provide comparisons for image reconstruction, semantic image synthesis, image captioning, and VQA in Table R1.1 below.  From these results, we can see that our method outperforms VQCT in all tasks, which suggests the effectiveness of our method. We also provide experimental results on integrating our method into VQCT further to verify the effectiveness and versatility of our method. Please refer to [the rebuttal pdf](https://openreview.net/attachment?id=QfryItnYya&name=pdf). We will include these experimental results and detailed discussion in our paper.
>
> **Table R1.1 Comparison of VQCT and our method on reconstruction, semantic synthesis, and visual text reasoning tasks.**
>
> | Model     | Codebook Size | #Tokens | CelebA-HQ | CUB-200  | MS-COCO  |
> | --------- | :-----------: | :-----: | :-------: | :------: | :------: |
> | VQCT      |     6207      |   512   |   5.02    | **2.13** |   9.82   |
> | VQ-GAN+LG |     1024      |   256   |   5.34    |   3.08   |  10.72   |
> | CVQ+LG    |     1024      |   256   | **4.90**  |   3.33   | **9.69** |
>
> |       | Semantic Synthesis on CelebA-HQ |                 | Image Captioning on CUB-200 |                  |                   |   VQA on MS-COCO   |
> | ----- | ------------------------------- | --------------- | --------------------------- | ---------------- | ----------------- | :----------------: |
> | Model | FID$\downarrow$                 | BLEU4$\uparrow$ | ROUGE-L$\uparrow$           | METEOR$\uparrow$ | CIDEr-D$\uparrow$ | Accuracy$\uparrow$ |
> | VQCT  | 14.47                           | 1.38            | 26.50                       | 24.63            | 98.22             |       40.42        |
> | Our   | **11.46**                       | **1.69**        | **34.73**                   | **25.78**        | **102.77**        |     **40.97**      |
>
>
>
>
>
> > Q2: Figs. 13 and 17 show image-to-image translation without language, while they're not compared with any other methods. The most basic unconditional image generation should be evaluated and compared.
>
> We appreciate your feedback to help us further improve the quality of our paper. Actually, we have provided the quantitative evaluation for Figure 13 in Table 5, row 2 of our original paper. Following your suggestions, we report the results for Figure 17, as shown in Table R1.2 below. Additionally, we conduct the unconditional image generation in Table R1.3 below. From the results, we can observe that our method achieves the best performance, which suggests our method's effectiveness. Moreover, we also provide some generation examples in Figure A of [the rebuttal pdf](https://openreview.net/attachment?id=QfryItnYya&name=pdf). We will include these experimental results in our paper.
>
> **Table R1.2 Comparison of image completion on CelebA-HQ.**
>
> | Model     | FID$\downarrow$ |
> | --------- | :-------------: |
> | VQ-GAN    |      9.02       |
> | VQ-GAN+LG |      8.14       |
>
> **Table R1.3 Comparison of unconditional image synthesis on CelebA-HQ.**
>
> | Model         | FID$\downarrow$ |
> | ------------- | :-------------: |
> | Style ALAE[2] |      19.2       |
> | DC-VAE[3]     |      15.8       |
> | VQ-GAN     |      10.2       |
> | VQ-GAN+LG     |       9.1       |
>
>
>
> > Q3: The proposed method requires the training images to have gt captions. How do you generate captions for image-only datasets?
>
> This is a good question. One possible way is to use existing pretrained visual-language models (VLMs) (e.g., MiniGPT-v2 [4], or ShareGPT4V [5]) to generate image captions. More importantly, by employing specific prompt strategies with these VLMs, we can obtain more detailed captions that contain richer and more useful knowledge. We believe that exploring this approach may further improve VQ-based model. We are very interested in exploring it in future work.
>
>
>
> > Q4: Is there a chance that the proposed method can be applied to other 2-stage generation frameworks, such as text-to-image latent diffusion's (especially the VQ-regularized) VAE?
>
> Yes, our method can be applied to other 2-stage generation frameworks. To verify this, following LDM [6], we use our method to conduct a text-to-image generation on the CUB-200 dataset. The experimental results are shown in Table R1.4 below. From the results, we can see that our method can further improve performance, verifying the versatility and effectiveness of our method. We also provide a qualitative comparison in Figure B of [the rebuttal pdf](https://openreview.net/attachment?id=QfryItnYya&name=pdf).
>
> **Table R1.4 Comparison of text-to-image on CUB-200 based on LDM.**
>
> |         | Text-to-Image on CUB-200 |
> | ------- | :----------------------: |
> | Methods |     FID$\downarrow$      |
> | LDM     |          35.13           |
> | LDM+LG  |          34.75           |
>
>
>
> ------
>
> [1] Zhang B, et al. Codebook Transfer with Part-of-Speech for Vector-Quantized Image Modeling. CVPR 2024.
>
> [2] Parmar G, et al. Dual contradistinctive generative autoencoder. CVPR 2021.
>
> [3] Pidhorskyi S, et al. Adversarial latent autoencoders CVPR 2020.
>
> [4] Chen J, et al. Minigpt-v2. arXiv 2023.
>
> [5] Chen L, et al. Sharegpt4v: Improving large multi-modal models with better captions. arXiv 2023.
>
> [6] Rombach R, et al. High-resolution image synthesis with latent diffusion models. CVPR 2022.

---

> > ### Comment · Reviewer_8ZRy · 2024-08-12
> >
> > Thank the authors for the detailed information. I think most of my concerns have been well addressed. Overall I'd like to keep my leaning toward acceptance.
> >
> > I'm having one more question regarding my Q3: I was thinking that you've generated captions for the image-only datasets mentioned in the paper, but in fact you didn't, did you? I'm wondering how the image-only datasets including CelebA-HQ, CUB-200 and MS-COCO were utilized in the experiments.

---

> > > ### Author Response · Authors · 2024-08-12
> > > **Official Comment by Authors**
> > >
> > > We are thankful for the reviewer’s response and feedback.
> > >
> > > > About Q3 image captions:
> > >
> > > In our paper, **we did not generate captions**.
> > >
> > > For CelebA-HQ, CUB-200, MS-COCO datasets, we **used publicly available image captions**, such as: CelebA-HQ from [1], CUB-200 from [2], MS-COCO from [3]. We will add these discussions to our final revised version.
> > >
> > >
> > > ------
> > >
> > > [1] Xia W, Yang Y, Xue J H, et al. Tedigan: Text-guided diverse face image generation and manipulation[C]//Proceedings of the IEEE/CVF conference on computer vision and pattern recognition. 2021: 2256-2265.
> > >
> > > [2] Reed S, Akata Z, Lee H, et al. Learning deep representations of fine-grained visual descriptions[C]//Proceedings of the IEEE conference on computer vision and pattern recognition. 2016: 49-58.
> > >
> > > [3] Chen X, Fang H, Lin T Y, et al. Microsoft coco captions: Data collection and evaluation server[J]. arXiv preprint arXiv:1504.00325, 2015.

---

> > > > ### Comment · Reviewer_8ZRy · 2024-08-13
> > > >
> > > > Thanks for the clarification. It's clear now. Please also add this information in the revised paper (the original section of datasets at Ln271 is over concise).

---

> > > > > ### Author Response · Authors · 2024-08-13
> > > > > **Official Comment by Authors**
> > > > >
> > > > > Thank you for your response. We will follow your comments and add these discussions to our revised version.

---

### Author Rebuttal · Authors · 2024-08-06

## Global Response

We sincerely thank all the reviewers for the thorough reviews and valuable feedback and will incorporate their suggestions into our next revision. We are glad to hear that the motivation is interesting (Reviewer YeAE), the paper is well-written (Reviewer YeAE) and easy to follow (Reviewer heW4, YeAE, and WVeZ), the proposed method is general and can be easily applied to VQ-based models (Reviewer 8ZRy, heW4, and WVeZ).

We have attempted to address all concerns, which has significantly improved the manuscript. We here summarize and highlight our responses to the reviewers:

* We introduce a novel work VQCT, recently published in CVPR 2024 and utilizing additional language and pretrained CLIP models. We provide detailed experimental comparisons on various tasks to address Reviewer 8ZRy's concern (unfair comparison) and Reviewer heW4's concern (the performance improvement comes from the image-text knowledge of the pre-trained CLIP model rather than our method).
* We provide experimental comparisons on many non-language image generation tasks and other 2-stage generation framework (Reviewer 8ZRy), experimental comparisons with other models in image generation and VQA (Reviewer heW4), experimental comparisons on visual grounding task (Reviewer YeAE), and ablation study on $L_{ras}$, to show the effectiveness of our method.
* We provide detailed analysis and discussion on Reviewer 8ZRy‘s’ question how do you generate captions for image-only datasets? We think this could be an exciting future work that will further advance the VQ-based community. We also add more detailed explanation about Reviewer heW4's questions (novelty and performance improvement) Reviewer YeAE's questions (Figure 1 and additional computational overhead) and Reviewer WVeZ's questions (the relationship alignment module and Figure 3).
* We provide some examples and experimental comparisons in the rebuttal pdf.

We reply to each reviewer's concerns in detail below their reviews. Please kindly check out them. Thank you and please feel free to ask any further questions. Finally, we feel this paper makes an important, helpful contribution (i.e, addressing the lack of multimodal information in current codebooks). Our method can significantly improve the performance of existing VQ-based methods in codebook representation, reconstruction, and various downstream tasks.

---

### Decision · Program_Chairs · 2024-09-25

**Decision:**

Accept (poster)

**Comment:**

The reviewers collectively appreciate the novel approach of integrating language guidance into codebook learning to enhance multimodal tasks, recognizing the method's effectiveness across various applications. While concerns about comparison fairness, clarity in specific module descriptions, and marginal performance improvements were raised, the reviewers acknowledge that these issues have been mostly resolved by the rebuttal. And in summary, the overall consensus is that the paper is well-written, the method is innovative, and the results are promising.  Based on these strengths, I recommend accepting the paper.